# Personality Determinants of Exercise-Related Nutritional Behaviours among Polish Team Sport Athletes

**DOI:** 10.3390/ijerph20054025

**Published:** 2023-02-24

**Authors:** Maria Gacek, Agnieszka Wojtowicz, Adam Popek

**Affiliations:** 1Department of Sports Medicine and Human Nutrition, Faculty of Biomedical Sciences, University of Physical Education in Kraków, 31-571 Kraków, Poland; 2Department of Psychology, Faculty of Social Sciences, University of Physical Education in Kraków, 31-571 Kraków, Poland; 3Bronisław Markiewicz State Higher School of Technology and Economics in Jarosław, 37-500 Jarosław, Poland

**Keywords:** athletes, Big Five, peri-exercise nutrition, personality traits

## Abstract

A proper diet increases the effectiveness of training and accelerates post-workout regeneration. One of the factors determining eating behaviour are personality traits, including those included in the Big Five model, i.e., neuroticism, extraversion, openness, agreeableness, and conscientiousness. The aim of this study was to analyse the personality determinants of peri-exercise nutritional behaviours among an elite group of Polish athletes practicing team sports. The study was conducted in a group of 213 athletes, using the author’s validated questionnaire of exercise-related nutrition behaviours and the NEO-PI-R (Neuroticism Extraversion Openness-Personality Inventory-Revised). A statistical analysis was performed using Pearson’s linear correlation and Spearman’s rank correlation coefficients as well as a multiple regression analysis, assuming a significance level of α = 0.05. It has been shown that the level of the overall index regarding normal peri-exercise eating behaviours decreased with increasing neuroticism (r = −0.18) and agreeableness (r = −0.18). An analysis of the relationship between the personality traits (sub-scales) of the Big Five model demonstrated that the overall index of proper peri-exercise nutrition decreased with the intensification of three neuroticism traits, i.e., hostility/anger (R = −0.20), impulsiveness/immoderation (R = −0.18), and vulnerability to stress/learned helplessness (R = −0.19), and four traits of agreeableness, i.e., straightforwardness/morality (R = −0.17), compliance/cooperation (R = −0.19), modesty (R = −0.14), and tendermindedness/sympathy (R = −0.15) (*p* < 0.05). A multiple regression analysis exhibited that the full model consisting of all the analysed personality traits explained 99% of the variance concerning the level of the proper peri-exercise nutrition index. In conclusion, the index of proper nutrition under conditions of physical effort decreases along with the intensification of neuroticism and agreeableness among Polish athletes professionally practicing team sports.

## 1. Introduction

Proper nutrition is an important factor determining exercise capacity and the effectiveness of post-exercise restitution processes [1,2,3,4,5]. They concern the time, quantity, and type of meals, snacks, and liquids consumed before, during, and after physical exercise, taking the specificity of the discipline and individual pre-dispositions as well as food preferences of the competitor into account [5]. Nutrition before training or a competition should be focused on adequate hydration and nutrition, during exercise, on replenishing fluids and energy losses, and after exercise, on accelerating post-exercise regeneration. A greatly significant aspect of peri-exercise nutrition is proper hydration, which is achieved by consuming water and isotonic drinks [1,2,3,4,5,6,7,8,9]. Pre-workout meals should be rich in carbohydrates (with different glycaemic indices) and low-fat protein products, as well as vitamins and mineral salts [5]. Before prolonged exercise (> 60 min), an additional energy reservoir may come from a carbohydrate snack [10]. Nutrition during post-exercise recovery should help restore disturbed homeostasis, optimise water and electrolyte balance, and aid the resynthesis of muscle and liver glycogen, while managing the acid-based balance and replenishment of cellular protein losses [1,2,3,4,5,11]. Indicators of nutrition and hydration status are among the significant biomarkers related to the health, performance, and post-exercise regeneration of athletes [12].

Meanwhile, in research among athletes, numerous quantitative and qualitative nutritional irregularities have been indicated. In this regard, a low supply of carbohydrates, vitamins (including antioxidants), and mineral salts (including potassium and calcium) has been found [13,14,15,16,17]. These are ingredients that play a key role in the energy of physical exercise, skeletal muscle contraction activity, and reduction of oxidative stress [1,10,18,19]. The described nutritional deficiencies may be associated with the insufficient consumption of products that have a high nutritional value, which has been indicated among athletes at research centres in different countries [20,21,22,23,24,25,26].

In the past few years, the health and nutritional behaviour of various population groups, including athletes, have been negatively affected by the COVID-19 pandemic [27,28,29,30]. At the same time, health training and a varied, balanced diet, rich in, among others, vegetables, fruits, and fish, containing immunostimulating ingredients (e.g., vitamins C and D and omega 3 PUFAs), can support the immune system and reduce health risks [29,31]. In endurance athletes, the relationship has been described between a rational diet, physical activity, and an improvement of physical capacity as well as body composition after a mild COVID-19 infection [32].

The nutritional behaviour of athletes is dynamic and conditioned by numerous factors, including personality [33,34,35]. Personality is one of the important aspects of human functioning in personal and social dimensions, related to, among others, cognitive, emotional processes, motivation, undertaken tasks, and achieving success. Personality determines the consistency of predispositions, mental functions, and the behaviour of individuals [36,37]. One of the dominant personality models in the psychology of traits is the Big Five model created by Costa and McCrae, which includes five main personality dimensions (neuroticism, extraversion, openness to experience, agreeableness, and conscientiousness), and their sub-categories [38,39]. Neuroticism describes a person’s level of emotional stability and resilience. People who score high in this dimension are sensitive and more frequently experience negative emotions, such as fear, anger or sadness, while people with low neuroticism are self-confident and emotionally stable. Extraversion refers to a person’s level of sociability, enthusiasm, and assertiveness. People who score high in this dimension tend to be outgoing, talkative, and energetic, while low scorers tend to be more reserved and introverted. Openness to experience refers to a person’s level of curiosity, creativity, and willingness to experiment. High scorers in this dimension are creative, open-minded, and interested in new experiences, while low scorers are more conventional and practical. Agreeableness is related to a person’s level of kindness, compromise, and empathy. People who score high on this dimension tend to be friendly, compassionate, and cooperative, while people who score low in this dimension tend to be more competitive and suspicious. Conscientiousness refers to a person’s level of self-discipline and responsibility. People who score high in this dimension tend to be responsible, effective, and goal-focused, while people with low scores tend to be more easy-going and less organised. In this way, the Big Five personality model allows for multi-faceted personality characteristics and explains socially and culturally significant behaviours that depend on the configuration of several personality traits at the same time [36,37]. Due to the significance of personality for success in sports, the personality assessment of athletes is an important area of sport psychology [40]. The level of neuroticism, extraversion, agreeableness, and conscientiousness can affect the results of competition in individual sports, although there is no single universal personality profile of athletes [40]. In studies among athletes, low neuroticism has mostly been noted, especially in high-level athletes [41,42,43,44].

Previous Polish research on the relationships between personality traits of the Big Five model and nutritional behaviours among people performing increased physical activity primarily concerned diet health quality among physical education students [45] and diet quality as well as nutritional behaviours among team sports athletes [46,47]. In the cited studies, the authors indicated relationships between the personality dimensions of the Big Five model and indicators of a healthy and unhealthy diet, implementing the qualitative recommendations of the Swiss nutrition pyramid for male athletes practicing team sports. The results of the above-mentioned studies mostly indicate the positive predictive significance of extraversion and conscientiousness, as well as the negative significance of neuroticism for the quality of athletes’ diets [46,47]. Relationships between personality traits and eating behaviours as well as nutritional status have also been the subject of research in population groups other than athletes [48,49,50,51].

To the authors’ knowledge, there is no research on the personality determinants of specific nutritional behaviours among athletes in conditions of physical exertion and post-exercise recovery. Therefore, due to the importance of diet in the peri-exercise period for the capacity of regeneration processes and the effectiveness of these processes, assuming the complexity regarding determinants of nutritional behaviours, a study was carried out on the personality determinants of athletes’ peri-exercise nutritional behaviours. The aim of this research was to analyse the personality determinants of peri-exercise nutritional behaviour among an elite group of Polish athletes professionally training in team sports. 

The following research questions were posed: (1) How are athletes’ peri-exercise nutritional behaviours shaped? (2) What are the relationships between personality traits and athletes’ peri-exercise nutritional behaviours?

Referring to the results of previous research [45,46,47] and the characteristics regarding the personality dimensions of the Big Five model (including neuroticism, associated with emotional liability, extraversion, regarding positive emotionality, conscientiousness, associated with the ability to control stimulus and being focused on achieving specific goals, and agreeableness, connected with less involvement in performed tasks) [36], a research hypothesis was formulated. It was assumed that personality traits are related to peri-exercise eating behaviours. Along with an increase in the level of extraversion and conscientiousness, the scale of correct eating behaviours also increases, and with the intensification of neuroticism and agreeableness, it experienced a decrease.

## 2. Materials and Methods 

### 2.1. Participants 

The research was carried out among a group of 213 Polish athletes (males) professionally practicing team sports, including basketball (n = 54), volleyball (n = 53), football (n = 53), and handball (n = 53). The basic criterion for selection into the study group was practicing sports at a professional level—at the level of the highest league in Poland, and for at least 3 years. The basic criteria for exclusion were belonging to the lower league class and/or failure to meet the criterion of minimum sports experience (3 years). The studied athletes, in relation to the current classification of the level of activity and sports abilities [52], can be assigned to Tier 3 (highly trained/national level). The age of the examined athletes was between 18 and 38 (M = 26.1; SD = 4.5), with the sports experience ranging from 3 to 20 years (M = 8.2; SD = 4.5). The median number of training sessions per week was 7, and the volume of a single training unit was 90 min. The study was performed in accordance with the principles of the Declaration of Helsinki, after obtaining informed consent from the participants. The research protocol was approved by the Bioethics Committee at the District Medical Chamber in Kraków (No. 105/KBL/OIL/2021).

### 2.2. Instruments

#### 2.2.1. Evaluation of Athletes’ Peri-Exercise Nutritional Behaviour

An original questionnaire regarding qualitative recommendations for peri-exercise nutrition was used to assess the nutritional behaviour of athletes. The questionnaire consists of 15 statements (items) concerning eating behaviours during the peri-exercise period. The responses were evaluated on the 5-point Likert scale (from 1 to 5, from “definitely no”, “rather no”, “hard to say”, and “rather yes” to “definitely yes”). The items included in the questionnaire concerned eating behaviours that are particularly important for post-exercise nutrition strategies, which increase the ability to exercise and the pace of regeneration processes, indicated by the authors of scientific papers in the field of nutritional recommendations for athletes [2,4,5]. The questionnaire enquiries concerned the following: intake of isotonic drinks during exercise, type of meal consumed before and after training, consumption of snacks and the type as well as amount of beverage intake before and after training, including drinks containing carbohydrates and electrolytes, as well as consumption of carbohydrate and protein products after training/competition. The subject of assessment was the athletes’ peri-exercise eating habits (during the previous 6 months). Based on the results of the questionnaire, the degree of implementing individual nutrition recommendations and the overall index of rational nutrition behaviours during the peri-exercise period were assessed (on a scale of 1–75 points, assuming that the higher the index, the more intense the rational peri-exercise eating behaviours). The questionnaire was validated. Test validity was assessed by repeated testing (n = 32). The value of the linear correlation coefficient was calculated and the H_0_ null hypothesis was tested: r = 0, via the Student’s *t*-test, obtaining a result confirming reliability of the scale (r = 0.378; *p* = 0.035). Good internal consistency of the scale was also confirmed (Cronbach’s α coefficient was 0.77).

#### 2.2.2. Evaluation of Athletes’ Personality Traits

The NEO-PI-R (Neuroticism Extraversion Openness-Personality Inventory-Revised) by P.T. Costa and R.R. McCrae [39] was used in the Polish adaptation by J. Siuta [53]. Characteristics of the NEO-PI-R personality inventory, according to the authors of the original tool and its Polish adaptation [39,53], have been presented in our previous publication [46]. Similarly, the personality traits of the examined group of athletes have already been the subject of our other publication [46]; therefore, they will not be presented in this work.

### 2.3. Statistical Analysis

The collected numerical material was subjected to statistical analysis using the Statistica 13.3 package. Statistical analysis was performed using Pearson’s linear correlation and Spearman’s rank correlation coefficients (depending on the nature of the variables). Multiple regression analysis was also carried out to check which of the variables could explain the level of proper peri-exercise index of nutrition. The stepwise progressive regression procedure (without intercept) was used in the calculations. The analysis also included the calculation of the multivariate determination coefficient (R^2^) and the standard error of estimation (s_y_), as well as the values of standardised partial regression coefficients b*, which are a measure of the relative significance regarding individual personality traits (independent variables X) in the model. The analyses were conducted assuming the significance level of α = 0.05.

## 3. Results 

### 3.1. Athletes’ Peri-Exercise Nutritional Behaviour 

With regard to implementing the recommendations of peri-exercise nutrition, it was found that almost all athletes (approx. 98%) consumed 200–250 mL of isotonic drinks after training. A high percentage (over 80%) consumed fruit and vegetables in their meals before and after training. At the same time, over 70% of the athletes consumed complex carbohydrates in the meal prior to training, 500–600 mL of fluids 2–3 h before training and carbohydrate products after exercise. More than half of the athletes declared the consumption of 1 litre of fluids per 1 h of training and a carbohydrate snack before long-duration training. To a lesser extent (about one-third of the group), the athletes consumed a snack at least 40 min pre-training, a meal at least 2 h before training, 200–600 mL of fluids immediately before training, and complete protein in their pre-exercise meals (Table 1).

The assessment of the peri-exercise eating behaviours (according to median) confirms that, to a high degree, the athletes consumed at least 1 litre of fluids per hour of training (Me = 4), complex carbohydrates in the pre-training meal, vegetables and fruits before training, 500–600 mL of fluids 2–3 h before training, a snack before training lasting more than 2 h, a meal within 30–60 min after training, and carbohydrates in the post-workout meal. Other assessed nutritional recommendations were implemented to a lesser extent and at a similar level (Me = 3.00). The overall index of proper peri-exercise nutrition was 51.9 points (out of 75 max) (Table 2).

### 3.2. Personality Traits and Peri-Exercise Nutritional Behaviour of Athletes

An analysis of the relationship between personality traits and the implementation of peri-exercise nutrition recommendations among athletes showed that the level of the overall index regarding correct eating behaviours (consistent with the recommendations of post-exercise nutrition strategies) decreased with increasing neuroticism (r = −0.18) and agreeableness (r = −0.18). In terms of particular aspects of peri-exercise nutrition, it was shown that with the intensification of neuroticism, the consumption of complex carbohydrates in the pre-workout meal (R = −0.15), snacks before more than 2 h training (R = −0.21), and complete protein consumption (R = −0.20) as well as complex carbohydrates in the post-workout meal decreased (R = −0.14). At the same time, with the intensification of extraversion, the consumption of at least 1 litre of water/isotonic drink for each hour of training decreased (R = −0.17) as well as the consumption of a meal within 30–60 min after ending training (R = −0.15), while the consumption of carbohydrates in the post-workout meal increased (R = 0.17). There was also a positive correlation between openness to experience and eating a snack before long-duration training (R = 0.17). Simultaneously, along with the intensification of agreeableness, the scale of consuming vegetables and fruits in the pre-training meal (R = −0.17), drinking 500–600 mL of fluids 2–3 h before training (R = −0.14), consuming carbohydrates in the post-workout meal (R = −0.21), and the intake of an isotonic drink in the amount of 200–250 mL every 15–20 min after training experienced a decrease (R = −0.14) (Table 3).

An analysis of the correlations between the personality traits (sub-scales) of the Big Five model showed that the overall index of proper peri-exercise nutrition decreased with the intensification of three neuroticism traits, i.e., hostility/anger (R = −0.20), impulsiveness/immoderation (R = −0.18), and vulnerability to stress/fear/learned helplessness (R = −0.19) and four traits of agreeableness, i.e., straightforwardness/morality (R = −0.17), compliance/cooperation (R = −0.19), modesty (R = −0.14), and tendermindedness/sympathy (R = −0.15) (*p* < 0.05) (Table 4).

A multiple regression analysis (dependent variable: overall index of proper peri-exercise nutrition; predictors: personality traits of the Big Five model) indicated that the full model consisting of all analysed personality traits explained 99% of the variance in the level of the index regarding appropriate peri-exercise nutrition, with agreeableness, extraversion, conscientiousness, and openness. The variable with the highest importance was agreeableness (b* = 0.437). The described correlations were directly proportional (Table 5).

## 4. Discussion

In the discussed research, limited implementation has been shown regarding qualitative recommendations for peri-exercise nutrition and significant correlations between some dimensions of personality and peri-exercise nutritional behaviours among elite Polish athletes practicing team sports.

When discussing peri-exercise nutrition, the average level of correct behaviours in this area should be highlighted (51.9 out of 75 points, i.e., 68.5%) and the varied level of implementing individual recommendations, including the highest (more than 70% of the group) regarding fluid replenishment before and after exercise, as well as vegetables, fruits, and complex carbohydrates in the pre-workout and post-workout meal. Among the recommendations of peri-exercise nutrition, special importance should be emphasized for supplementing water and electrolytes as well as vegetables and fruits (alkalinizing products, which are, among others, a source of antioxidants, B vitamins, magnesium, potassium, and carbohydrates). This is also true for other carbohydrate products in restoring homeostasis and the optimisation of post-exercise restitution processes, that is, restoring the water–electrolyte and acid-base balance and rebuilding carbohydrate losses (accelerating muscle and liver glycogen resynthesis processes) [1,2,3,4,5]. Consuming appropriate amounts of vegetables and fruits (rich in antioxidant substances, including polyphenols, vitamin C, and carotenoids) contributes to the reduction of oxidative stress indices, i.e., reducing the health risks associated with its level [1,54,55,56,57]. In a situation of high exposure to oxidative stress, in conditions of vigorous physical exercise, a diet rich in dietary antioxidants (including vegetables and fruits) is an important aspect of rational nutrition for athletes [55,58,59,60].

The significance of proper fluid replenishment (and prevention of dehydration in sports) is emphasized by numerous authors [8,9,61,62]. In the research on the subject, the use of isotonic drinks has also been indicated in the effective replenishment of water and electrolyte loss among athletes, including those training volleyball, American football, and rowing [63,64,65]. The subject of numerous studies in the field of sports dietetics was also the assessment of carbohydrate supply as the basic energy substrate in the diet of athletes. In recent meta-analytical studies, the prevalence of low energy and carbohydrate intake among team sports athletes has been confirmed [17]. In other trials, the occurrence has been noted with regard to qualitative nutritional irregularities among athletes, including those practicing team sports [20,21,22,23,25].

Before discussing the relationships between personality traits and exercise-related eating behaviours, it is necessary to point out the basic personality characteristics of the athletes under study. In this regard, it was found that the athletes obtained high scores for extraversion (M = 121.8), openness (M = 115.0), agreeableness (M = 123.2), and conscientiousness (M = 128.5), while low scores were observed for neuroticism (M = 72.1) [46]. A low level of neuroticism among athletes has also been described in other studies on professional athletes, including those practicing team sports [41,42], also among those from Poland [43], and especially among master-class athletes [44].

The discussed research allowed us to note statistically significant correlations between the personality traits of the Big Five model and their sub-scales and the quality of peri-exercise nutrition among athletes. A negative predictive value regarding neuroticism and agreeableness was found in the overall index of proper peri-exercise nutrition. The correlations found between extraversion and exercise-related eating behaviours were not unambiguous, while within the dimension of openness, a positive relationship was described with one of the aspects of nutrition, i.e., the snack before long-duration training. Conscientiousness and its sub-scales were not related to the quality of peri-exercise nutrition among the studied athletes. The obtained results confirm the difficulties in an unambiguous assessment and interpretation of the relationship between the personality and nutritional behaviours of athletes. 

The discussed study Is a continuation of our earlier research that was carried out among Polish team sports athletes, which concerned correlations between personality traits and the health quality of the diet (associated with the frequency of consuming products with potentially beneficial and potentially adverse health effects) and with the implementation of the quality recommendations from the Swiss pyramid for athletes [46,47]. The discussed results, indicating the negative predictive significance of neuroticism (and its sub-scales) for normal exercise-related nutritional behaviours, refer to the relationship between neuroticism and a lower health quality of athletes’ diets [46]. Furthermore, among physical education students, the relationship between lower neuroticism and more rational food choices in terms of consuming sea fish was described [45]. In other studies conducted among the general population, it was shown that neuroticism, through the mechanism of emotional eating, promoted the consumption of non-recommended products, including confectionery [48]. The discussed studies on athletes, indicating the negative predictive significance of agreeableness (and its sub-scales), correspond to studies among university students in Ghana, in which a relationship was demonstrated between high agreeableness and irregular eating habits [49]. Agreeableness reduces commitment to performed activities. The described ambiguous relationships between extraversion and some particular exercise-related eating behaviours of athletes training team sports correspond to the positive predictive significance of extraversion as an indicator of a healthy diet of athletes shown in our previous research [46], but also as an indicator of healthy and unhealthy diets among students of physical education [45]. On the one hand, higher extraversion favoured the consumption of vegetables among athletes [46], but also, confectionery products among physical education students [45]. Relationships between conscientiousness and the quality of peri-exercise nutrition were not described, unlike among students of physical education (in them, along with the increase in conscientiousness, the pro-health quality of the diet, expressed by the pro-healthy diet index, pHDI-14, increased) [45]. Positive relationships between conscientiousness and a healthy diet were also noted by other authors in various population groups other than athletes [48,50,51].

It can be concluded that various studies on the personality determinants of eating behaviour in different population groups sometimes provide varied and ambiguous results. Further interdisciplinary research is needed to explain the mechanisms of the observed relationships, which is also pointed out by other authors [51]. Nutritional irregularities found among athletes justify the need to monitor diet and carry out nutritional education, considering the individualisation of influences promoting a healthy way of eating, also in conditions of physical exercise and post-exercise recovery. Learning about the relationships between personality and eating behaviours may be conducive to the personalisation of interactions in the field of nutrition education and diet modification, taking the personality traits of athletes into account. By understanding how personality traits are related to nutrition, we can better identify individuals who may be at a higher risk of poor health outcomes and consequently develop targeted interventions to promote healthy eating. As we learn more about the interplay between personality and nutrition, we may be able to develop more personalised approaches to nutrition. For example, individuals who demonstrate a high level of neuroticism may benefit more from different types of dietary interventions than individuals who exhibit high extraversion. By tailoring our recommendations to an individual’s personality, we may be able to achieve better outcomes.

The limitations of this work are primarily related to the failure to include demographic and sports variables (see training, competition practice, and discipline), as well as one selected nutritional area (peri-exercise nutrition behaviours) and the self-descriptive nature of the applied research tools. The limitations of the study also concern the failure to consider the training loads that determine the nutritional needs of athletes. The limitations indicated as well as others may set the directions for further research, the aim and subject of which should be to achieve a comprehensive assessment of personality determinants in various areas of sports nutrition, taking gender, sports experience, sports level, and type of discipline into account. Further research could concern personality determinants regarding the quantitative aspects of athletes’ diets (e.g., energy consumption, macronutrients, vitamins, and mineral salts), which would contribute to a comprehensive assessment of athletes’ nutrition.

## 5. Conclusions 

Athletes practicing team sports showed an average level of implementing qualitative recommendations for peri-exercise nutrition (approx. 68%). The highest percentage of the studied athletes correctly supplemented fluids and also consumed vegetables and fruits as well as complex carbohydrates in pre- and post-workout meals, while in a smaller percentage, they consumed, e.g., complete proteins in pre-workout meals.Significant correlations were found between the personality traits of the Big Five model and the quality of peri-exercise nutrition among athletes, with the overall index of proper nutrition under conditions of physical effort decreasing along with the intensification of neuroticism and its three sub-scales (hostility/aggression, impulsiveness/immoderation, and vulnerability to stress/fear/learned helplessness) as well as agreeableness and its four sub-scales (straightforwardness/morality, compliance/cooperation, modesty, and tendermindedness/sympathy). Thus, the negative predictive significance was confirmed of neuroticism and agreeableness for the quality of the athletes’ peri-exercise dietary choices.The obtained results indicate the need to monitor the diet and nutritional education of athletes in the area of recommendations for peri-exercise nutrition, taking personality traits into account, as well as the need for further research in order to explain the mechanisms of the observed relationships between personality traits and an athlete’s diet.

## Figures and Tables

**Table 1 ijerph-20-04025-t001:** Implementing recommendations of peri-exercise nutrition among athletes training team sports (percentage of indications).

Nutritional Behaviours	1 Definitely No	2 Rather No	3 Difficult to Say	4 Rather Yes	5 Definitely Yes	No 1 + 2	Yes 4 + 5
At least 1 litre of water/isotonic drink/1 h of training	0	2.3	35.7	40.4	21.6	2.3	62.0
Main meal at least 2 h before training	2.3	14.6	46.9	28.2	8	16.9	36.2
Complex carbohydrates in pre-training meal	0	2.3	20.2	46.9	30.5	2.3	77.4
Complete protein in pre-training meal	4.2	9.9	51.6	26.8	7.5	14.1	34.3
Vegetables and fruits in pre-training meal	0	6.1	13.1	50.2	30.5	6.1	80.7
500–600 mL of fluids 2–3 h before training	0.5	1.4	23	46	29.1	1.9	75.1
200–600 mL of fluids immediately before training	5.6	12.2	46	22.5	13.6	17.8	36.1
Snack before above 2 h training	3.3	8	35.2	36.2	17.4	11.3	53.6
Snack at least 40 min before training	4.7	11.3	45.5	25.8	12.7	16.0	38.5
Snack before training in liquid or semi-liquid form	1.9	13.1	42.3	26.3	16.4	15.0	42.7
Meal within 30–60 min after training	0	2.8	16.4	50.2	30.5	2.8	80.7
Complete protein in post-training meal	3.3	14.1	54.5	17.4	10.8	17.4	28.2
Carbohydrates in post-training meal	0	9.4	17.8	42.3	30.5	9.4	72.8
200–250 mL of isotonic drink after training every 15–20 min	0	0	2.3	19.7	77.9	0.0	97.6

Legend: 1—definitely no, 2—rather no, 3—difficult to say, 4—rather yes, and 5—definitely yes.

**Table 2 ijerph-20-04025-t002:** Implementing recommendations of peri-exercise nutrition among athletes training team sports (descriptive statistics).

Nutritional Behaviours	M	SD	Min	Max	Me	Q25	Q75
At least 1 litre of water/isotonic drink/1 h of training	3.81	0.80	2.00	5.00	4.00	3.00	4.00
Main meal at least 2 h before training	3.25	0.88	1.00	5.00	3.00	3.00	4.00
Complex carbohydrates in pre-training meal	4.06	0.77	2.00	5.00	4.00	4.00	5.00
Complete protein in pre-training meal	3.23	0.89	1.00	5.00	3.00	3.00	4.00
Vegetables and fruits in pre-training meal	4.05	0.83	2.00	5.00	4.00	4.00	5.00
500–600 mL of fluids 2–3 h before training	4.02	0.79	1.00	5.00	4.00	4.00	5.00
200–600 mL of fluids immediately before training	3.26	1.03	1.00	5.00	3.00	3.00	4.00
Snack before above 2 h training	3.56	0.98	1.00	5.00	4.00	3.00	4.00
Snack at least 40 min before training	3.31	0.99	1.00	5.00	3.00	3.00	4.00
Snack before training in liquid or semi-liquid form	3.42	0.98	1.00	5.00	3.00	3.00	4.00
Meal within 30–60 min after training	4.08	0.76	2.00	5.00	4.00	4.00	5.00
Complete protein in post-training meal	3.18	0.92	1.00	5.00	3.00	3.00	4.00
Carbohydrates in post-training meal	3.94	0.93	2.00	5.00	4.00	3.00	5.00
Overall index of proper peri-exercise nutrition	51.9	3.85	39.0	63.0	49.0	52.0	54.0

Legend: M—arithmetic mean, SD—standard deviation, Me—median, Q25—lower quartile, and Q75—upper quartile.

**Table 3 ijerph-20-04025-t003:** Correlations between personality traits and implementing peri-exercise nutrition recommendations among athletes training team sports (Pearson’s r and Spearman’s R correlation coefficients).

Nutritional Behaviours	N	E	O	A	C
	Pearson’s r				
Overall index of correct peri-exercise eating behaviours (total from items)	−0.18 *	0.01	−0.01	−0.18 *	0.06
	**Spearman’s R**				
At least 1 litre of water/isotonic drink/1 h of training	−0.02	−0.17 *	−0.07	0.08	0.01
Main meal at least 2 h before training	0.03	−0.02	−0.03	−0.04	−0.02
Complex carbohydrates in pre-training meal	−0.15 *	−0.10	0.02	−0.01	0.06
Complete protein in pre-training meal	0.11	−0.04	0.01	0.01	0.05
Vegetables and fruits in pre-training meal	−0.13	0.10	0.12	−0.17 *	0.03
500–600 mL of fluids 2–3 h before training	−0.02	0.07	−0.02	−0.14 *	0.01
200–600 mL of fluids immediately before training	−0.07	0.01	−0.03	−0.09	0.09
Snack before above 2 h training	−0.21 *	0.02	0.17 *	−0.02	0.10
Snack at least 40 min before training	0.01	−0.05	−0.06	0.01	−0.03
Snack before training in liquid or semi-liquid form	−0.01	0.03	−0.01	−0.07	−0.03
Meal within 30–60 min after training	−0.12	−0.15 *	−0.02	−0.06	0.03
Complete protein in post-training meal	−0.20 *	0.08	0.12	−0.06	0.06
Carbohydrates in post-training meal	−0.14 *	0.17 *	0.07	−0.21 *	0.02
200–250 mL of isotonic drink after training every 15–20 min	0.10	0.06	−0.13	−0.14 *	−0.01

N—neuroticism, E—extraversion, O—openness, A—agreeability, and C—conscientiousness. ** p* < 0.05.

**Table 4 ijerph-20-04025-t004:** Correlations between the characteristics (sub-scales) of the personality dimensions from the Big Five model and the overall index of proper peri-exercise nutrition among athletes training team sports (N = 213) (Spearman’s rank correlation coefficient values).

Main Dimensions of Personality	Characteristics—Sub-Scales of Personality Dimensions	Overall Index of Proper Peri-Exercise Nutrition (Spearman’s R)
Neuroticism	N1—anxiety	−0.07
	N2—hostility/anger	−0.20 *
	N3—depression	−0.12
	N4—impulsiveness/immoderation	−0.18 *
	N5—vulnerability to stress/fear/learned helplessness	−0.19 *
	N6—self-consciousness	−0.11
Extraversion	E1—gregariousness/sociability	−0.01
	E2—warmth/kindness	−0.02
	E3—assertiveness	0.12
	E4—activity level/lively temperament	0.02
	E5—excitement seeking	−0.09
	E6—positive emotion	−0.02
Openness to experience	O1—fantasy/imagination	0.04
	O2—aesthetics/artistic interest	0.06
	O3—feelings/emotionality	−0.02
	O4—action/adventurousness/exploration	0.06
	O5—ideas/intellectual interest/curiosity	−0.09
	O6—values/psychological liberalism/tolerance to ambiguity	−0.04
Agreeableness	A1—trust—in others	−0.04
	A2—straightforwardness/morality	−0.17 *
	A3—altruism	−0.10
	A4—compliance/cooperation	−0.19 *
	A5—modesty	−0.14 *
	A6—tendermindedness/sympathy	−0.15 *
Conscientiousness	C1—competence/self-efficacy	0.03
	C2—order(lines)/organising	0.06
	C3—dutifulness/sense of duty/obligation	0.06
	C4—achievement striving	0.10
	C5—self-discipline/willpower	0.11
	C6—deliberation/consciousness	0.06

* *p* < 0.05.

**Table 5 ijerph-20-04025-t005:** Correlations between personality traits in the Big Five model and the overall index of proper peri-exercise nutrition among athletes training team sports (multiple regression analysis).

Personality Traits	R^2^	S_y_	b*	Std. Error with b*	B	Std. Error with b
Agreeableness	0.990	5.192	0.437	0.943	0.184	0.018
Extraversion			0.340	0.052	0.144	0.022
Conscientiousness			0.134	0.041	0.053	0.016
Openness			0.091	0.064	0.041	0.029

R^2^—coefficient of multivariate determination; S_y_—standard error of estimation; and b*—standardised partial regression coefficient.

## Data Availability

Not applicable.

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
