# Peer review of "Personality Determinants of Exercise-Related Nutritional Behaviours among Polish Team Sport Athletes"

_ijerph, 2023, doi:10.3390/ijerph20054025_

Round 1

Reviewer 1 Report

Lines 53 to 67 and 139 to 145 have a different format. In table 2, the presence of the dividing lines between each nutritional behavior is more of a nuisance than an aid to reading. They should be eliminated or perhaps just leave a thinner one, as in table 3. In table 4, the lines that are found differentiating the subscales should be eliminated; the separation lines of Main dimensions of personality are sufficient. Line 279.- The comment "(however, these were mostly weak)" is not necessary. Line 278.- the parenthesis should be eliminated. Line 284-285.- the parenthesis should be eliminated. Line 288-289.- the parenthesis should be eliminated. Line 290-292.- Parentheses are used to provide a comment that is totally dispensable but helps in some way to remember something or to think about something. In the case of this sentence it should be eliminated and considered as a contribution to the discussion, that is, as a new sentence associated with the idea previously exposed with a connector. Line 295.- The phrase in parentheses should be changed to "...as it is the snack before long-duration training, e.g." Line 295.- Delete the parenthesis "... Conscientiousness and its sub-scales were..." Line 300-301.- Delete parenthesis Line 344.- Change "i.e." by "e.g".

Reviewer 2 Report

The conclusion part in the abstract needs modification to present the message from the current study.

In the introduction, it is better to discuss about the effect of adherence to high-quality dietary pattern on COVID-19 outcomes and also the role of baseline physical activity on mortality and disease outcomes in COVID-19.

In the discussion, ‘’ In the discussed research, an average level has been shown regarding the implementation of qualitative recommendations for peri-exercise nutrition and significant 237 correlations between some dimensions of personality and nutritional behaviour in conditions of physical exertion and post-exercise recovery among elite Polish athletes training team sports.’’ Please revise this part and present your intent clearly.

In the discussion, nearly all parts start with this sentence: ‘’The discussed research’’; please revise them accordingly.

The final conclusion needs modification to present the message from the current study.

Reviewer 3 Report

I commend the authors for examining personality determinants of peri-exercise nutritional behaviors in Polish team sport athletes. Your study is interesting, however, there are several areas that need to be addressed and clarified. See specific comments below.

Abstract:

The big-five model should be mentioned earlier in the abstract.

Introduction:

L36: Define peri-exercise nutrition.

L38: Add reference

L39: What do is considered prolonged exercise? Please define.

L45: Clarify what is meant by “mistakes have been indicated”. Mistakes by whom? Also, mistakes is strong language.

L47: Demonstrated by whom? From the sentence starting on L 45.

L 53: “One of the most important” – this is a strong statement.

L 58: I suggest stating Big Five Personality here and then staying consistent throughout the manuscript. Also, it would be helpful to define these variables.

L60: Elaborate on the subcategories. Provide a few examples.

L65-67: Be careful since there is no one specific personality profile for athletes. I suggest softening the language.

L68: What championship level? What does this mean?

L81: Perhaps change psychological determinants to personality determinants.

L89: I am not sure the current study addresses research question 1.

L97-100: The hypothesis needs to be clarified. What is meant by higher, more rational food choices vs less beneficial ones?

Overall, I want more connection on why it is important to understand the connection between personality and nutrition.

Materials and Methods:

L 118: Is there a reference for the peri-exercise nutritional behavior instrument? Need more information on the questionnaire. What are the directions and the stem for the items? Is this in general, during the last training session? It is not fully clearly what exactly is being measured? How were the variables (e.g., complex carbs) explained to the athletes?

How was demographic information collected?

Was training load taken into consideration with the nutritional demands?

Results:

L161: The recommendations of peri-exercise nutrition should be mentioned in the introduction. However, this will vary depending on the training loads.

Table 1: I suggest writing the responses instead of 1-5 in the first row. This will make it easier to read. Same with table 2.

L 191: How are you defining “correct eating behaviors”.

Discussion:

L 271-273: I recommend that you report in the results section the personality characteristics of the participants.

L277-301: This is re-stating the results without any connection to previous literature or why these findings are important. Perhaps combine with the paragraph below and add in additional information.

L302-303: Since this is a continuation, this should be mentioned in the intro.

L304: Define what is meant by health quality of the dieter?

L309: Again, what is considered rational food choices?

L333-334: Nutritional mistake is a strong statement that is used throughout.

L337-340 Limitations: A major limitation to the study is not considering the training loads of the athletes. As it is mentioned, individualizing nutrition based on these needs is necessary. Age and sport information is included in the demographics. Why were analyzes not carried out on these variables?

I would like to see more information on the practical application of this study. If I were working with a team sport, what should I do based on this study? Yes, I should provide nutritional education and monitor athletes but, how is this different in terms of personality?

Overall, the article should be revised for grammatical errors. The writing style was difficult to follow. Check the paper for differences in spacing.

Round 2

Reviewer 2 Report

none

Reviewer 3 Report

Thank you for revising the manuscript. All changes have been made.